# OpenReview forum: "Particles Don’t Care About Z: Towards Scaling Entropy Estimation of Unnormalized Densities"
_ICML.cc/2026/Conference — ICML 2026 regular_

### Official Review · Reviewer_dmum · 2026-03-06

**Soundness:** 3
**Presentation:** 1
**Significance:** 3
**Originality:** 2
**Overall Recommendation:** 3
**Confidence:** 4

**Summary:**

The paper proposes a parameters selection method and Metropolization of Stein variational gradient descent. For the Metropolis-Hastings steps the authors require the density of the inferred distribution. To compute this density, they rely on a previous paper (P-SVGD) which derives a formula via the residual representation and Banachs fixed point theorem. The authors improve this density estimator and put some effort in selecting the step sizes and bandwidth of the underlying kernel.

**Compliance With Llm Reviewing Policy:**

Affirmed.

**Final Justification:**

After the rebuttal my main concerns remain:

1. The presentation remains chaotic, the figure captions are too small to be read, the template instructions remain violated and the text is not straightforward to read.

2. There is no real comparison with other methods despite P-SVGD. In the rebuttal the authors compare to a figure which isn't even placed in the numerical part but referred as some kind of motivation in the introduction. As far as I understand from the reply the authors main argument is their emphasize on estimating the entropy instead of just generating samples. However, also for SMC or diffusion-based samplers can evaluate the energy (there were at least three papers at any of the recent major ML conferences). The considered GMM problems are quite easy and most recent samplers handle these kind of problems close to perfection. In order to be accepted, I think the authors should include a

3. Even after the authors reply I am not really convinced from backpropagating through a whole sampler which sounds incredibly expansive (even with the approximations proposed by the authors (and others before in an optimization context))

In summary, I think there is some material there and the approach is worth trying. But the way how it is presented, placed in the literature and compared to prior work is not convincing. Therefore I still vote weak reject.

**Key Questions For Authors:**

- In which sense is the derived density estimate better than using a kernel density estimator on the inferred partilces?

- Kernel methods (and SVGD) use to be sensitive to the number of considered particles. How is it chosen in the experiments and how do the requirements with the number of particles change with the dimension?

**Limitations:**

The authors don't explicitly discuss limitations. A few directly come into my mind:

- Quadratic complexity in the number of particles

- poor scaling of kernel methods with the dimension (I am not sure whether this is partially solved by the band-width selection, but I haven't found a discussion of this issue)

- potentially accumulating errors in the density estimation steps

**Strengths And Weaknesses:**

I have some concerns regarding the presentation:

- The writing is a bit chaotic (the figures are too full and refer to too many methods which are not explained in the paper and are not always properly explained or referenced in the text), but with careful reading it can be figured out what is meant by the authors.

- Formatting: The paper violates the ICML style guide at several points. Specifically, the abstract is too long, the running title is not adapted (and states "Submission and Formatting Instructions for ICML 2026") and parts of the paper have a reduced space between the line (e.g. line 90 to 103 (left), 181 to 191 (left), 220 to 228 (right), 265 to 274 (left) etc.). Also the space around the section titles is reduced and figure titles are extremely long making the paper looking a bit clumpsy.

From a methodogical viewpoint, the paper proposes several small improvements of the P-SVGD algorithm. None of these results is really new, but their combination leads to a fair contribution. Some comments:

- The parameters are learned by an algorithmic unrolling approach. That is, the authors run the algorithm until some approximate equilibrium. If I understand Alg 1 in the Appendix correctly, the authors propose to backpropagate through the whole chain produced by SVGD, which looks like it could be incredible expansive (in terms of memory and in terms of computational cost). To address this the authors write that they "backpropagate only every k steps in large-scale experiments". Similar problems frequently appear in bilevel optimization (which is actually what the authors do without calling it like this) and several solutions were developed over time (like Jacobian-free backpropagation, or differentiating the fixed point via the implicit function theorem). The authors should discuss and relate their method to such approaches.

- The Metropolization of the gradient flow reminds a bit on previous papers [1,2,3] and some discussion of metropolized (importance reweighted) models should be included in the related work.

The experiments are a bit chaotic:

- I am missing an application to standard Benchmarks. E.g. take a couple of test distributions from the MCMC literature (e.g. selected from GMM40 in different dimensions, Funnel, Many Well) and include a table of the estimated entropy and, if applicable, some two-sample comparison between generated and ground truth particles. If this is done properly, the method can also be compared to the literature (either by the authors themselves, or by the readers) without reimplementing the method or the comparisons. So far, the authors only compare to SVGD based comparisons and plain Langevin sampling.

- The experimental setting is missing some structure. Please define properly the target distribution, the evaluation metrics and the comparisons. Interpretations and evaluations can be drawn after the setting is clear.

[1] "Sequential Monte Carlo samplers", Journal of the Royal Statistical Society B

[2] "Adaptive Monte Carlo augmented with normalizing flows", PNAS, 2022

[3] "Importance Corrected Neural JKO Sampling", ICML, 2025

---

> ### Author Rebuttal · Authors · 2026-03-31
>
> We thank the reviewer for the feedback.
> ____
> **Presentation.** We respectfully disagree that the writing is chaotic. We made a deliberate effort to structure the paper clearly, which is reflected in feedback from other reviewers _(e.g., Reviewer 7SgF: “The organization of the paper is really stellar.”)_.  We also believe that the methods presented in the figures are carefully explained in the text. We would appreciate more concrete pointers. That said, we acknowledge that the paper may appear dense in places. This is partly due to the abstract nature of the problem and our effort to provide precise formulations and detailed illustrations within a strict page limit. Importantly, the camera-ready version will include **an additional page**, which will allow us to further improve readability by introducing more spacing and increasing some figures' sizes.
> ____
> **Significance of the contribution.** We respectfully disagre that our work consists of incremental improvements to P-SVGD. MET-SVGD advances the long-standing problem of entropy estimation for unnormalized densities and achieves SOTA results. In particular:
> * **Theory.** We derive a unified bound linking invertibility and trace approximation, providing the first formal guarantee for this class of estimators. Incorporating an MH correction into inherently non-reversible SVGD dynamics is non-trivial and required a new construction, including a novel expression for the induced density and an efficient acceptance probability.
> * **Analysis.** We identify fundamental failure modes of P-SVGD in high dimensions (Fig.~2). While some components build on established techniques (e.g., adaptive step sizes, learned bandwidths), the key contribution lies in diagnosing these issues and proposing targeted principled fixes.
> * **Empirical gains.** MET-SVGD achieves up to $16\times$ higher accuracy than prior SVGD baselines and up to $64\times$ improvements in training stability on CIFAR-10 EBMs, along with an $80.4%$ improvement in FID. In maximum-entropy RL, it yields up to $16%$ higher returns, demonstrating substantial gains over existing methods.
> ___
> **Adaptive number of steps.** Thank you for the insightful pointers to alternative backprop approaches. In our setting, however, these techniques are not straightforward to apply due to the **non-stationary and stochastic nature of the SVGD–MH dynamics**: each step involves a **different mapping** as the learned parameters (e.g., kernel bandwidth) evolve over steps, and the inclusion of the MH correction further introduces stochasticity. The truncated algorithmic unrolling (backprop every $k$ steps) was manageable as we learn an expressive initial distribution, hence we didn’t need a large number of steps. We report the number of steps for MaxEntr RL in Fig. 42. We will revise the paper to explicitly discuss these alternative approaches and their trade-offs as promising directions for future work.
> ____
> **MH related work.** Thank you for the references. We'll add these papers to the related work.
> ____
> **MCMC benchmarks.** Our work is primarily positioned within the variational inference and SVGD literature, rather than the classical MH framework. Accordingly, our experiments focused on standard benchmarks in these settings, including Gaussian and GMMs (Figs. 2a, 2d, 6b, 17, and 23), as well as scalability benchmarks commonly used in the SVGD literature (Fig. 2d). Yet, we agree that including additional MCMC-style benchmarks would improve comparability with a broader range of methods. We'll incorporate these in the revised appendix.
> ___
> **Experimental setting.** The target distributions and evaluation metrics are already explicitly described in the main text, while all detailed experimental settings are provided in the appendix in 16 tables (Tabs. 4–19) and referenced in the paper. Due to space constraints, we aren't able to include all these details in the paper.
> ___
> **Kernel density estimators** can be fitted to the inferred particles, but they suffers from poor scalability in high dimensions and act only as a post-processing step that does not improve the sampling procedure itself. In contrast, our method learns a parametric representation of the distribution **jointly** with sampling, allowing the model to guide exploration and improve sample quality.
> ___
> **Sensitivity to the number of particles.** There is no practical rule for selecting the number of particles $M$ in SVGD: the best known rate (Shi \& Mackey, 2024) scales as $\mathcal{O}(1/\sqrt{\log \log M})$, which is too weak to guide practice. In line with prior work (including P-SVGD), we choose $M$ based on computational budget, using tens to a few hundred particles and increasing it moderately with dimensionality to ensure sufficient coverage. We report the exact numbers in the appendix.
> ___
> **Limitation section.** Kindly check our reply to Reviewer U1oC.
> ___
> We hope we have addressed all concerns and would be happy to clarify any remaining questions.

---

> > ### Author Rebuttal · Reviewer_dmum · 2026-04-01
> >
> > Thank you for your replies.
> >
> > Some of my concerns remain unaddressed:
> >
> > - Presentation: I think reviewer U1oC provided a good summary of the presentation issues. In addition, one major problem is that P-SVGD is discussed before it is introduced. Given the fundamental role of P-SVGD for this paper, consider to introduce the basic mechanisms of P-SVGD before discussing it's strengths or weaknesses. It is natural that the presentation of a paper is received differently across the reviewers, but reading the other reviews I feel that I am not alone with my concerns. Despite that there are a couple of clear format violations (reducing vertical spaces between the lines, undersized figures) and it appears a bit unfair to me to grant some authors more space than others (but I will ignore this for the moment; it is not up to me to judge this).
> >
> > - Experimental benchmark: There are plenty of sampling (and entropy estimation) methods in literature (in the MCMC literature, but also neural samplers became increasingly popular recently). While I agree with the authors that none of those "solves" the problem, it is highly unclear how MET-SVGD compares to them. The numerics mostly compares MET-SVGD with P-SVGD. However, the claim that MET-SVGD achieves SOTA results (as formulated by the authors) must be underlined with some evidence which I currently do not see.
> >
> > For the moment I will keep my score.

---

> > > ### Author Response · Authors · 2026-04-02
> > >
> > > We thank the reviewer for the follow-up and are glad that some concerns were addressed. We respond to the remaining points below.
> > >
> > > **P-SVGD is discussed before it is introduced.** We do introduce P-SVGD and its key contribution (the density formulation) early in the introduction (L. 99). Our intent was to keep the introduction focused and avoid overloading it with technical details which could hinder readability.
> > >
> > > That said, we do acknowledge that certain aspects, such as the invertibility assumption, the divergence control heuristic, and the Hessian term simplification, are first mentioned in the following limitations paragraph and only fully explained later (in the related work and method sections). To improve clarity, we will revise the introduction to briefly summarize these key assumptions and mechanisms upfront. Specifically, we will add the following clarifying sentence (Col. 2, L95, before “Under mild assumptions”):
> > >
> > > > "The derivation assumes invertibility of the SVGD dynamics and leverages a sampling-based approximation of expectations in Eq. 1 to avoid explicit Hessian computations. The method further incorporates a divergence control heuristic that truncates deviating particles, enabling strong empirical performance.”
> > > ____
> > > **Experimental benchmark.** Our evaluation targets benchmarks that are directly aligned with our problem setting: constructing variational distributions from unnormalized densities that (i) exploit the structure of the target, (ii) are sufficiently expressive, (iii) remain computationally tractable, and (iv) support efficient sampling (as stated in the abstract and illustrated in Fig. 1).
> > > Accordingly, we compare against methods that satisfy these criteria. In particular, since our approach builds on SVGD, we include **five state-of-the-art SVGD variants specifically designed to address scalability** (Fig. 2d), providing a strong and targeted comparison within the most relevant family of methods.
> > >
> > > Beyond SVGD, **normalizing flows are the only widely adopted class that provides tractable density estimates in this setting**, and we therefore include flow-based baselines in both domains: Glow-NF (EBMs, Fig. 7) and SAC-NF (RL). As discussed in the corresponding sections, these models consistently underperform in our regime, which is expected given the well-documented difficulty of training flows under reverse KL objectives. Importantly, the MH-based baselines suggested by the reviewer [2,3] are themselves flow-based and are only validated in low-dimensional settings. As such, they do not provide a competitive or scalable alternative in the regimes we consider. We believe this addresses the concern regarding comparisons to broader sampling techniques.
> > >
> > > In contrast, the beauty of both P-SVGD and MET-SVGD is that **they directly leverage the score of the target distribution within their dynamics and density estimation**. This turns out to be more powerful than relying on generic invertible transformations and leads to substantially more stable and scalable learning, particularly in high dimensions.
> > >
> > > Given this positioning and the breadth of comparisons included, we believe it is justified to claim state-of-the-art performance both in (i) entropy estimation from unnormalized densities within the considered setup and (ii) improving the scalability of SVGD, which is an active and independent research direction.
> > > ____
> > > We hope this addresses the review concerns, and we are happy to clarify any remaining questions.

---

### Official Review · Reviewer_7SgF · 2026-03-11

**Soundness:** 4
**Presentation:** 4
**Significance:** 4
**Originality:** 3
**Overall Recommendation:** 5
**Confidence:** 4

**Summary:**

This paper explores entropy estimation, density modeling, and sampling from an unnormalized probability distribution, using Stein Variational Gradient Descent (SVGD) methods.
They explore an elegant recently introduced method, P-SVGD, and systematically explore a series of algorithmic / implementation details that lead to poor performance in practice.
For each limitation, the paper derives a principled solution.
In experiments they show that they greatly improve entropy estimation accuracy over previous SVGD, as well as more stable sampling and improvement on a RL task.

**Compliance With Llm Reviewing Policy:**

Affirmed.

**Final Justification:**

I read responses and rebuttals for all reviewers, and felt the authors addressed concerns.

The generality of the framing (compared to p-svgd) and application (lots of work by authors to make this approach work in many scenarios, high-d, etc) led to an increase in significance score and overall score.

In the reviewer discussion after the rebuttal phase, there were compelling points that in some ways the scope of this work is limited, as there were no comparisons with other non-SVGD methods. That is not a problem, I think, but a score of 6 is really meant for flawless papers with a broad significance. I think to justify that we would have needed to see that P-SVGD also excels as a sampler, compared to a more broad class of methods. Still, this seems like solid work that should be accepted.

**Key Questions For Authors:**

The minor comments are just suggested for helping improve the paper. The only substantial question is if the authors can comment on whether MET-SVGD is, or could become, the SOTA across all types of methods for some tasks.

**Limitations:**

yes

**Strengths And Weaknesses:**

# Strengths

- The organization of the paper is really stellar. Fig. 2 visually expresses the limitations, Fig. 3 the solutions. There's a nice mapping between contributions and problems solved. (Though Fig. 2 is a bit dense - I'm not sure if it can be simplified though.)
- I had not encountered P-SVGD, but the intro to it in this paper was very intuitive, and it seems like an elegant approach that is worth perfecting.
- Lots of interesting ideas are employed, not just for show but in logical and effective ways. (Stuff like fixed point theorem, Stein violation, neat autograd tricks, HMC, MH)
- Complexity discussion for each addition is also nice.

# Weaknesses

This paper seemed like careful and high quality work. I think the only questions that can change the overall score are about significance. (a) Is it a major advance over P-SVGD? (b) Is the method "globally significant", i.e., does this look like a SOTA way to accomplish some task compared to any other possible method.

Personally, I find the results convincing that this is a major advance over P-SVGD, so I'll ask some questions about the overall significance of the approach. The results really only compare within the SVGD family.
I think the easiest way to demonstrate this method's broader significance would be to compare it to other classes of methods  (neural entropy estimators, flow or diffusion based approaches) and show that it is the best at something. I understand that most methods aren't suited for unnormalized densities. Still, I just reviewed a paper that follows up on a line of work doing this with diffusion models (A good ref seems to be
> Rissanen, S., OuYang, R., He, J., Chen, W., Heinonen, M., Solin, A., and Hern´andez-Lobato, J. M. Progressive tempering sampler with diffusion. arXiv preprint arXiv:2506.05231, 2025.

These are good for sampling but not as obviously for entropy estimation, so maybe there are no true comparisons. If so, then it should be emphasized more strongly.

Another broad question about the global significance has to do with my vague worries that the complexity of the method will make it fragile in real-world scenarios. Your experiments with CIFAR and RL somewhat mitigate this concern, but I still am dubious that any method that uses kernels can scale past a few thousand dimensions, even with adaptive step tricks.



## Minor comments:
I felt that the intro parts were a little repetitive in places. E.g. the intro to Sec. 3 kind of repeats the Intro, Sec. 1.

Are the new Hutchinson estimators very expensive? You say "Only one additional first-order derivative..." is required, but I didn't quite follow this as the term above seemed to have two derivatives. Maybe the first one is done with forward mode autograd, since it's a directional derivative.

Issues with Fig. 7? All the figures were actually on the small side. For a camera-ready, perhaps you can choose some results to enlarge and others for appendix. (Fig. 8 also had rendering issue for me, on y-axis label.)

I found the title amusing and memorable, maybe for people without physics or Stein background it's more opaque.

---

> ### Author Rebuttal · Authors · 2026-03-30
>
> We would like to thank the reviewer for such a supportive and engaged review. Your feedback was very insightful!
> ___
> **Global Significance and Positioning.** Our method addresses a fundamental challenge that is orthogonal to sampling: while diffusion-based models (e.g., diffusion models) excel at generation, they do not provide tractable density or entropy estimates from unnormalized distributions. To our knowledge, this work establishes the first accurate and scalable framework for entropy estimation from unnormalized densities in high-dimensional settings (Fig. 1). This problem is long-standing and central to key machine learning paradigms, including Maximum Entropy Reinforcement Learning and Energy-Based Models. While P-SVGD was an important initial step, our approach introduces a principled correction that significantly improves both accuracy and scalability, effectively establishing a new state-of-the-art for this under-explored problem. We will revise the abstract and introduction to more clearly highlight this positioning.
>
> [1] Rissanen, S., OuYang, R., He, J., Chen, W., Heinonen, M., Solin, A., and Hern´andez-Lobato, J. M. Progressive tempering sampler with diffusion. arXiv preprint arXiv:2506.05231, 2025.
> ___
> **Scalability.** We agree that in high-dimensional Euclidean spaces, selecting an appropriate kernel bandwidth is non-trivial, as both overly large and overly small values can lead to vanishing gradients. However, we argue that MET-SVGD is inherently robust to these challenges in practical settings where **data concentrates on low-dimensional manifolds**. Its scalability is supported by two key mechanisms:
> * Expressive Initial distributions: By backpropagating through the sampling chain, we learn parameters of expressive initial distributions that place the initial particles near the relevant manifold, significantly reducing the burden on the kernel to operate across large, poorly scaled distances
> * Step-wise adaptive kernel bandwidth:  as stated by the reviewer and demonstrated by our experiments, this has significantly helped SVGD scalability.
>
> In the future, we intend to experiment with kernel decomposition techniques, which can further help scalability.
> ___
> **Repetition Sec. 3/ Sec. 1.** The brief recap at the beginning of Sec. 3 was intended to improve readability by restating the problem and contributions supported by Fig. 1 before introducing the technical details. We acknowledge that it may come across as repetitive, we will streamline the introduction paragraph to introduce the contributions at a higher level.
> ___
> **Hutchinson estimator complexity.** Only an additional backpropagation (the outer derivative) is introduced as:
> * The inner derivative (the score) is already computed as part of the SVGD update
> * A single random vector $v$ per particle is used, following Song et al., 2020 in their Sliced Score Matching work. In practice, the summation over particles within a batch provides a sufficient stochastic approximation of the trace and has been proven effective in high-dimensional deep learning.
> ___
> **Figure size and rendering.** We will leverage the **additional page** available in the camera-ready version to increase figure sizes and improve readability. Regarding Fig. 8, we have verified the rendering of the y-axis and do not observe any issue on our end; this may be due to a PDF viewer–specific artifact. We will continue investigating to ensure consistent rendering across platforms.
> ___
> **Title.** We are glad the reviewer found the title amusing and memorable. That was very much the intent! :)
>
> To ensure "opacity" doesn't turn into "obstruction," we will add a brief footnote (or a "For the non-physicist" sentence) clarifying that $Z$ is the normalization constant.
> ___
> We hope the above clarifications address the reviewer’s concerns regarding the significance of our work, and we would appreciate reconsideration of the score if the reviewer finds the responses satisfactory.

---

> > ### Author Rebuttal · Reviewer_7SgF · 2026-04-02
> >
> > Thanks for the response, I've also read the other reviews and rebuttals. Generally, the points raised seem largely cosmetic, and I'm satisfied with the suggested improvements. (And no need to explain the title, I think.)
> >
> > Sorry for being late to respond, and not leaving much time. I'd be willing to consider going to 6, the main question in that case is about the significance of the method which, to me, hinges on the scale of innovation over the (Messaoud, 2024) work, which I had not encountered before. The writing in this paper casts the improvements as relatively straightforward engineering improvements to improve speed and accuracy. But, glancing just now at the p-svgd, I was surprised that framing seems quite different. I'll try to study p-svgd a little more, but thought I'd comment in case the authors have more thoughts on this.

---

> > > ### Author Response · Authors · 2026-04-03
> > >
> > > Thank you very much for the thoughtful follow-up and for taking the time to look into P-SVGD. We truly appreciate it! We clarify the scale of innovation over P-SVGD in the following.
> > >
> > > The original work (Messaoud et al., 2024) was introduced in the context of maximum-entropy RL, where the goal was to learn expressive actor policies with tractable entropies in soft actor critics. The authors propose modeling the actor as an SVGD sampler and derive the density in Eq. 2 (our paper). We were intrigued by the potential of the approach as:
> > > * It can offer a highly expressive variational distribution for a **wide range of ML applications beyond RL**
> > >
> > > * The derived density leverages the target score directly, leading to more parameter-efficient models than generic invertible transformations in flow models
> > >
> > > * It provides a systematic way to learn sampler parameters, which is a long-standing goal in the sampling community
> > >
> > > However, P-SVGD lacked the **theoretical rigor** and **stability** required to serve as a general-purpose inference method. Our innovation lies in formalizing and stabilizing this approach, **transforming it from a task-specific RL heuristic into a robust, high-dimensional inference framework**. Specifically, we address four fundamental limitations that hinder its broader application:
> > >
> > > * **From informal local invertability heuristic to global theoretical guarantees.** The validity of the density in P-SVGD relies on the invertibility of the particle transformation. P-SVGD utilizes the Implicit Function Theorem, which only guarantees **local invertibility** under **loose, non-practical bounds**  ($\epsilon << \sigma$, Proposition 3.2, P-SVGD paper). This is compounded by **a second loose bound required for the trace approximation** (Theorem 3.1, P-SVGD paper). In practice, these constraints offer little guidance, forcing a reliance on "small enough" step sizes that unnecessarily hamper convergence. Our work replaces these heuristics with a **unified, exact, and efficient bound** that ensures **global invertibility and justifies the trace approximation**, providing a mathematically rigorous constraint that guarantees a well-defined density at every point in the flow (Sec. 3.1 in our paper).
> > >
> > > * **Resolving Hyperparameter Sensitivity.** P-SVGD noted high sensitivity to kernel variance but incorrectly attributed this to invertibility violations (Sec 4.1). Their suggested remedy, a "hand-wavy" recommendation to choose kernels that "guarantee inter-dependencies", provided no concrete implementation. We prove that **the instability is not an invertibility issue**, but rather the result of cumulative approximation noise across steps. We resolve this by (1) learning step-wise kernel bandwidths and step sizes (Sec 3.2), and (2) deriving an efficient stopping criterion based on Stein identity violation (Proposition 3.4.). This enables an adaptive number of steps, effectively transforming the method into a deep flow model with dynamic depth. This provides the stability necessary for precise variational inference, a level of reliability that P-SVGD lacked.
> > >
> > > * **Ensuring Correctness at Scale.**  P-SVGD was evaluated on low-dimensional RL tasks (max 17D). In scaling to thousands of dimensions, we identified two critical failures:
> > >    * Hessian Omission: We proved the omission of the trace of the Hessian is mathematically incorrect for finite particle sets. We introduced a Hutchinson-based estimator to correct this efficiently (Sec 3.3).
> > >    * Mode Collapse: We found their divergence control heuristic (particle truncation) actually exacerbates mode collapse. We replaced this with a Metropolis-Hastings (MH) augmentation that preserves the topology of the target distribution.
> > >
> > > * **From weak to strong convergence guarantees.** the MH step aslo led to strong to convergence guarantees.
> > >
> > > By resolving these foundational issues, we demonstrate that our framework achieves state-of-the-art results for entropy estimation from unnormalized densities, while also advancing the state-of-the-art in SVGD scalability to high-dimensional settings. We compare against P-SVGD, **five baselines from the SVGD scalability literature** (Fig. 2d, Fig. 6a, Fig. 23), and flow-based models (Figs. 7 and 8), which are the primary generative approaches that provide tractable density estimates and consistently achieve an order-of-magnitude improvement in performance across these benchmarks.
> > >
> > > We hope this helps clarify the scale of our contributions and will be helpful in your assessment of the work.

---

### Official Review · Reviewer_U1oC · 2026-03-12

**Soundness:** 2
**Presentation:** 1
**Significance:** 2
**Originality:** 2
**Overall Recommendation:** 3
**Confidence:** 3

**Summary:**

This paper propose the method MET-SVGD that mitigates the flaws of the previous method P-SVGD on differential entropy estimation. The main improvements are: MET-SVGD unifies the step-size constraints into a single principled one satisfying global invertibility instead of local invertibility like P-SVGD. MET-SVGD learns SVGD’s hyperparameters and eliminating the need for SVGD hyperparameter tuning in P-SVGD. MET-SVGD adaptively determines the number of steps of SVGD to guaranteee convergence. MET-SVGD restores the trace-of-Hessian term to scale up to high dimensions. MET-SVGD utilizes MH steps to avoid mode collapse problems. The paper experiments on multiple real-world tasks to show the effectiveness of the proposed method.

**Compliance With Llm Reviewing Policy:**

Affirmed.

**Final Justification:**

The authors partially addressed my concerns. However, I maintain the viewpoint that the current paper is flawed not only in its presentation but also in its unclear contribution justification and its narrow scope.

From the presentation perspective, I believe the writing improvements require a significant update to the paper.

From the contribution justification perspective, in subsection 3.4, the key advantages of the proposed method MET-SVGD could be more clearly presented with more solid theoretical evaluations. The authors' reply does not explicitly provide theoretical reasoning regarding the specific regularity level of the target distribution at which P-SVGD would fail while MET-SVGD would still succeed.

From the scope of the method comparison perspective, the authors do not provide specific numerical comparison results with other relevant methods such as [1]. I think that it would be beneficial to include a comprehensive discussion on the relevant methods (mentioned in the rebuttal and other reviewers) in the current paper to achieve a broader impact.

Therefore, I would like to keep my score.

[1] Cantwell, G. T. Approximate sampling and estimation of partition functions using neural networks. arXiv preprint arXiv:2209.10423, 2022.

**Key Questions For Authors:**

1. The paper claims in Table 1 that MET-SVGD inherits advantages of different approximate inference methods, including sample efficiency. Why is MET-SVGD sample efficient? In my view, the additional MH reduces the sample efficiency due to the rejection probability. Additionally, the memory complexity also increases due to deepnets.
2. Does the proposed method outperform other entropy estimation methods like [1] in experiments?
3. The paper claims that Figure 2d shows that P-SVGD limits scalability to high dimensions. However, in my view, the blue line of P-SVGD and the brown line of MET-SVGD does not differ from each other essentially. Could the authors explain a bit more on this issue?

[1] Cantwell, G. T. Approximate sampling and estimation of partition functions using neural networks. arXiv preprint arXiv:2209.10423, 2022.

**Limitations:**

The paper lacks more in-depth discussions on the possible limitations (like sample efficiency and complexity) of the proposed method MET-SVGD.

**Strengths And Weaknesses:**

Strengths:
1. The content of the paper is rich, including theoretical analysis and numerical experiment comparisons that support the claim of the paper.
2. The proposed method improves the previous algorithm P-SVGD from various aspects, supported with in-depth discussions.

Weaknesses:
1. The writing of this paper is difficult to read. The abstract is too long and some details could be omitted. The line spacing is too dense and the paragraphs could be divided into smaller paragraphs for a better focus. It would be better to use more displayed equations rather than inline equations for the ease of reading. The references (like in line 40 and line 173) should be arranged in chronological order. The proof sketches are too simple to understand the gist of the theoretical analysis. The graphs of the paper are too small to read. In addition, the word choice and sentence construction could be improved (like avoiding stating a single sentence “This is incorrect” at line 220). I would suggest rearranging the current version of the paper, reserving the most important points (like the improvement of Metropolis-Hastings) in the main body, moving minor details to the appendix.
2. The paper mostly focuses on comparing with a single method P-SVGD and mostly addresses the limitations of P-SVGD without discussing the possible limitations of the proposed method MET-SVGD, narrowing the scope of the paper.
3. The exact convergence rate improvement of MH on the bounds of SVGD convergence in the finite-particle setup is not explicitly presented, weakening the theoretical support for the additional MH design of the proposed method. It would be beneficial to evaluate more on the theoretical improvement of MH in non-smooth targets tasks to better support the claim at line 327: "addressing the limitation of P-SVGD 'prevents divergence but does not improve convergence to non-smooth targets'".

---

> ### Author Rebuttal · Authors · 2026-03-30
>
> We thank the reviewer for the positive comments on the richness of our paper.
> ____
> **[W1] Formatting.** While we acknowledge that the paper may appear dense in places, this is partly due to the abstract nature of the problem and our effort to make the contributions concrete through detailed illustrations and precise formulations under page limit. Importantly, the camera-ready version includes **an additional page**, which will allow introducing more spacing, converting inline equations into displayed ones, and increasing some figures' sizes. Nevertheless, we note that we received positive comments on organization and readability from other reviewers (e.g., **Reviewer 7SgF: “The organization of the paper is really stellar”**).
> ___
> **[W1] Chronological order of the references (L40-173).** Thanks, we’ll adjust.
> ___
> **[W1] Proof sketches (Prop. 3.1 and Cor. 3.3).** The full proofs are technically involved and consist of multiple steps, requiring substantial space to be presented in detail. Thus, we focus on conveying the main ideas in the proof sketches and refer the reader to the appendix for complete derivations.
> ___
> **[W1] Word choice and sentence improvement.** The sentence "his is incorrect" (L. 20) was used deliberately to emphasize what we consider a central issue in prior work, namely the misdiagnosis of invertibility in P-SVGD. That said, we are happy to soften the phrasing to improve the tone. More generally, we have carefully revised the manuscript for clarity and would appreciate specific examples where the wording could be improved, so we can address the concern more concretely.
> ___
> **[W2] MET-SVGD limitations.** We discuss limitations in Sec. 4.2 and Sec. 4.3, and will add a dedicated section summarizing the following:
> * The entropy estimate based on the derived density (Eq. 2) introduces a stochastic error whose variance is of the order $\mathcal{O}(\sqrt{d/M})$ with $d$ being the dimensionality. In principle, this error may accumulate over iterations. In practice, our framework mitigates this by learning expressive initial distributions, which reduces the number of required steps to convergence (see Fig. 42).
> * The RBF kernel complexity is quadratic in the number of particles. In the future, we'll experiment with efficient approximations like random Fourier features or Nyström ones, which complexity is linear in $M$.
> * The MH step requires an average of $\mathcal{O}(L_c B / 2)$ evaluations of the target score, as described in L350-356, where $B$ denotes the maximum number of reverse steps (set to 10 in our experiments).
> ___
> **[W3] Deriving the convergence rates improvements induced by the MH correction in the finite-particle regime** is technically intricate and constitutes a substantial future research problem on its own. We do not believe that the absence of explicit rate guarantees weakens the support for our method. Our contributions focus on diagnosing key limitations of P-SVGD, providing corrected derivations, and introducing a principled MH correction, all of which are supported both theoretically and empirically.
> ___
> **[W3] Convergence proof for non-smooth targets (L 327).** No additional proofs are required for this claim, as the MH correction introduces well-established convergence guarantees that apply to general target distributions, including non-smooth ones. As a result, MET-SVGD addresses the limitation of P-SVGD in L 327.
> ___
> **[Q1] Sample efficiency.** In MET-SVGD, rejected particles **are not discarded**; they remain in the particle set and can be updated in subsequent iterations (see L.165-166, Col. 2). Thus, MH doesn’t reduce sample efficiency.
> ___
> **[Q1] Memory complexity** is negligible as the additional step-size and kernel bandwidth deepnets are lightweight. The architectures are in Tabs. 4, 14, 15, 18 and 19 with all less than 10 fully connected layers.
> ___
> **[Q2] Comparing to [1].** This method targets partition function estimation via learning a normalizing flow, which is known to be challenging to train. Also, experiments only cover low-dimensional setups. In general, approximating the partition function requires solving a high-dimensional integral, which is fundamentally harder and less scalable than expectation estimation, as in our approach.
> ___
> **[Q3] Fig. 2d.** at $d=1000$, P-SVGD exhibits an error of approximately 130 in entropy (~10\% relative error), which is substantial given that the target is a high-dimensional Gaussian. In contrast, MET-SVGD closely tracks the ground truth across dimensions.
> ___
> **Overall complexity.** Complexity of the Invertibility condition, optimized SVGD parameters and corrected derivation are listed in  L260-263,  L302-313, L311-313, respectively and are not significant as the trace term is heavily reused and the deepnets are lightweight. The main additional cost arises from the MH step as explained above.
> ___
> We believe these clarifications address the reviewer's concerns and kindly ask for a score reconsideration.

---

> > ### Author Rebuttal · Reviewer_U1oC · 2026-04-02
> >
> > Thanks for the detailed response. Though it helps me understand the paper better, some of my concerns remain unaddressed.
> >
> > 1. On the presentation and the writing, I maintain my concern on the clarity of the paper. In my view, the writing of the paper requires major revision. The proof sketches are too vague for the readers due to only providing well-known facts and statements without linking closely to the Proposition and Corollary needs to be proved. For example, on the proof sketch for Corollary 3.3, it would be beneficial to explicitly show that $A=\nabla_{x^l}\phi(x^l)$ rather than using a general letter $A$. I believe if reserving the most important points of the current paper in the main body and moving relatively minor details to the appendix, there will be more space to provide more informative proof sketches and larger graphs, etc. Revising the writing to be cleaner and clearer will help the readers leave a deeper impression on the improvement of the methods that the authors wish to convey. I respectfully disagree with the "additional page" reply as it may introduce unfairness.
> >
> > 2. On W3, the authors replied with "No additional proofs are required for this claim, as the MH correction introduces well-established convergence guarantees that apply to general target distributions". I would like to see more concrete and detailed theoretical evaluations on "to what extent" smoothness of the target distribution will the P-SVGD fail while MET-SVGD still works. It would be beneficial for authors to present theoretical reasonings (or at least cite relevant references that presents the "well-established convergence guarantees") including the smoothness level (Lipschitz constant/conditioning number) of the target. I believe the wording "non-smooth distribution" at the first row of Table 6 in the Appendix H.1.2 that gives the implementation details for the "non-smooth target" visualization Fig. 2b is inaccurate (since it is actually smooth) and should be change to something like "large conditioning number of the variance matrix" describing that the target is close to a singular distribution.
> >
> > 3. On Q2, it would still be informative to quantitatively compare with [1] on low dimensional setups. This will broaden the scope of the paper. The authors replied "This method targets partition function estimation via learning a normalizing flow", but [1] actually shows "how variational autoencoders (VAEs) can be applied to this task". The networks for VAE "are considerably less constrained" (cited from [1]) and may be easier to train, which is left to be examined by quantitative experiments. I am also not exactly convinced about the statement "normalizing flows are the only widely adopted class that provides tractable density estimates in this setting" in the "Reply Rebuttal Comment" for Reviewer dmum, since there may be other classes of methods like VAE. It would be beneficial to include a comprehensive discussion on these relevant methods in the paper.
> >
> > I would like to keep my score.

---

> > > ### Author Response · Authors · 2026-04-03
> > >
> > > Thank you for following up. We are glad that we were to address some of the concerns and hope to address the rest in the following.
> > >
> > > **Presentation.** We understand the reviewer’s concern regarding formatting and fairness. Our intention was not to gain additional advantage, but to present the material as clearly as possible, given the technical density of the contribution.
> > > Importantly, the raised formatting concerns are limited (e.g., figure sizes and spacing) and do not affect the correctness, reproducibility, or interpretation of the results. More broadly, we believe the primary focus of the review process should be on the novelty, soundness, and significance of the work, while presentation aspects, although important, will be systematically improved in the camera-ready version. We will ensure full compliance with formatting guidelines in the revision.
> > > ____
> > > **Proof sketches.** we thank the reviewer for the constructive feedback and revise both sketches to include more details:
> > >    * **Prop. 3.1.**  To recover the inverse $x$ of $f(x) = x + \epsilon \phi(x)$, we consider the fixed-point iteration
> > > $x = y - \epsilon \phi(x)$, where $y = f(x).$ The existence and uniqueness of this fixed point (i.e., the inverse) follow from the Banach fixed-point theorem, which requires the mapping $x \mapsto y - \epsilon \phi(x)$ to be contractive. This holds when $\phi$ is Lipschitz and $\mathrm{Lip}(\epsilon \phi) < 1$, where the Lipschitz constant is given by
> > > $\mathrm{Lip}(\phi) = \sup_{x \neq y} \frac{\|\phi(x) - \phi(y)\|_2}{\|x - y\|_2} \le \sup_x \|\nabla \phi(x)\|_2,$ with the inequality following from the mean value theorem.
> > >
> > >  * **Corr. 3.3.** We leverage the inequalities $|\lambda_{\max}(A)| \le \|A\|_2 \le \sqrt{\mathrm{Tr}(A A^\top)},  \forall A \in \mathbb{R}^{d \times d}$.  Specifically, for  $A = \nabla \phi(x)$, the conditions
> > >     * $\epsilon |\lambda_{\max}(\nabla \phi(x))| < 1$  (Proposition 3.2) and
> > >     * $\epsilon \sup_x \|\nabla \phi(x)\|_2 < 1$ (Proposition 3.1) can be unified into the sufficient condition $ \epsilon \sup_x \sqrt{\mathrm{Tr}(\nabla \phi(x)\nabla \phi(x)^\top)} < 1$
> > >
> > > ______
> > > **Advantage over P-SVGD in non-smooth target setups.**
> > >    * Terminology - Smoothness vs. Conditioning: We acknowledge that "non-smooth" can be interpreted in multiple ways. While we used the term to describe targets with large Lipschitz constants (large gradients), we agree that in the context of Gaussian targets (Fig. 2b), "co-variance with large conditioning number" is a more precise. We will add it to the caption as an explanation for smoothness.
> > >    * Why P-SVGD fails on non-smooth targets: The fundamental issue with P-SVGD in these settings is its sensitivity to the interplay between the drift and repulsion terms. Specifically: in regions of high gradients (large Lipschitz constants), the drift term is strong. When combined with the heuristic median kernel bandwidth, the resulting particle updates often "overshoot" the target manifold, resulting in poor convergence as shown in Fig. 2b. P-SVGD lacks a mechanism to control or correct such updates. The divergence control heuristic only removes particles that deviate significantly, rather than preventing these unstable dynamics.
> > >    * Theoretical Guarantees of MET-SVGD: By augmenting SVGD with an MH rejection step, we move from the weak convergence of standard SVGD (minimizing KL divergence) to the strong convergence of MCMC. Unlike SVGD, the MH-step ensures that the stationary distribution is exactly the target, regardless of the target’s smoothness or conditioning, provided the chain is irreducible and aperiodic (satisfied by our construction). The samples converge **pointwise** (every sample with probability 1) to the target distribution. As noted in (Mengersen & Tweedie, 1996) (cited at L. 346), these strong guarantees **do not require smoothness assumptions on the target**, making the approach significantly more robust.
> > > _____
> > > **Comparison to VAEs and to [1].** We would like to clarify that standard VAEs optimize a variational lower bound (ELBO) and do not provide a tractable expression of the data density in general. In particular, evaluating the likelihood requires integrating over latent variables, which is intractable without additional approximations. As such, they are not directly comparable to our setting, where we explicitly require generative models with a tractable density and entropy.
> > >
> > > Regarding [1], while it is framed in terms of VAEs, it relies on learning a transformation (decoder) that is effectively a normalizing flows model. In practice, this introduces comparable training challenges, especially in high dimensions. We implemented [1] and will include the results in the paper. Empirically, we observe behavior similar to flow-based models (e.g., Glow-NF in Fig. 7 and SAC-NF in Fig. 8, including instability during training, which limits performance in our setting.

---

### Official Review · Reviewer_aqPx · 2026-03-13

**Soundness:** 3
**Presentation:** 3
**Significance:** 3
**Originality:** 3
**Overall Recommendation:** 4
**Confidence:** 4

**Summary:**

In this paper, the authors develop MET-SVGD, a Stein variational gradient descent based methodology to approximate the differential entropy associated to an unnormalised density. In particular, the authors base their algorithm on previous work (P-SVGD), but suggest a number of theoretical and methodological improvements.

**Compliance With Llm Reviewing Policy:**

Affirmed.

**Key Questions For Authors:**

It would be good to report the overall number of score evaluations and compare to P-SVGD. Also, it would be interesting to know which of your modifications you think are most important for performance. Would it be possible to test them separately?

**Limitations:**

yes

**Strengths And Weaknesses:**

The paper is empirically/methodologically strong, likely providing real practical value for the development of SVGD and related algorithms.

Soundness: The diagnosis of problems in P-SVGD seems good, and the modifications (restored trace/Hessian correction, learned bandwidths and step sizes, adaptive stopping) are plausible and supported by experiments. My main concern is about the Metropolis-Hastings step: since the authors do not implement or provide the exact accept/reject ratio (there is an error term in Proposition 3.5), saying that SVGD "inherits asymptotic convergence guarantees from MH, [...] converges strongly to the target" (around line 345, right-hand column) seems like an overstatement. Also, the algorithm seems fairly expensive because of the training phase, so it would be valuable to report the required number of target score evaluations in the experiments.

Presentation: The presentation is ok, but the general flow could be improved and some parts could be more polished. Some minor points:

(i) The optimal transport reference (Villani, 2009) is cited in various places which seem odd (for example on variational inference).

(ii) Proposition 3.4: the statement is unclear on its own, because Stein's identity violation is not defined.

Significance: The contribution is likely useful for other researchers working on similar algorithms based on interacting particles.

Originality: The diagnosis of problems in P-SVGD and the modifications are original and interesting.

---

> ### Author Rebuttal · Authors · 2026-03-30
>
> We thank the reviewer for the positive feedback on the methodological and empirical strength of our approach and address the remaining concerns in the following:
> ___________________
> **[MH step] accept/reject ratio.** The error term in Proposition 3.5 is explicitly controlled: Corollary 3.3 enforces a truncation on the step size $(\epsilon^l < \epsilon_{UB}^l, \forall l \in [0,L-1])$ which ensures that the approximation error term in Proposition 3.5 remains strictly bounded, i.e., $\frac{\epsilon^l}{ \epsilon_{UB}^l} <1$. Besides, since the error term $\mathcal{O}\big(\big(\frac{\epsilon^l}{ \epsilon_{UB}^l}\big)^2\big) $ scales **quadratically** with the step size, the resulting bias in the acceptance ratio becomes negligible. Consequently, it is reasonable to state that SVGD inherits asymptotic convergence guarantees from MH up to a controlled vanishing approximation error. We will clarify this point in the paper.
> _______________________
> **[MH step] Training complexity**. As stated in L350-356 (Col. 2), the MH step introduces $\mathcal{O}(L_c B / 2)$ additional backpropagations in expectation, stemming from the reverse SVGD updates computed via the Banach fixed-point theorem (Alg.~3), where $B$ denotes the maximum number of reverse steps (set to 10 in our experiments).
> _______________________
> **The optimal transport reference (Villani, 2009)** is cited when discussing the expressivity of the SVGD density estimate (Eq. 2), namely its ability to approximate a broad class of target distributions. This property follows from optimal transport theory: the result holds for distributions that can be obtained via smooth transformations of a tractable reference distribution. We will clarify this point to make the connection more explicit.
> _______________________
> **Stein Identity violation.** We'll revise Proposition 3.4 to clarify: **Stein's identity** is defined as the condition $SI(q^l, p) = 0$, as stated in Appendix A.6. By **violation of Stein's identity**, we refer to the deviation from this condition, i.e., $SI(q^l, p) \neq 0$.
> _______________________
> **Ablation study and relative importance of contributions.** We provide an extensive ablation study of the different components in Fig. 7 and Fig. 8. Based on these results, we can rank the main contributions in terms of their impact on performance:
>
> 1. _Corrected derivation (trace term + step size upper bound)_. Restoring the trace-of-Hessian term (Sec. 3.3) consistently and significantly improves performance. In addition, enforcing the step size upper bound $\epsilon_{UB}$ (Sec. 3.1), which ensures invertibility of the transport map, is critical. Omitting either component leads to clear performance degradation (see w/o $\mathrm{Tr}\nabla^2$ and w/o $\epsilon_{UB}$ in Fig. 7 and Fig. 8).
>
> 2. _Learning the kernel bandwidth (Sec. 3.2 )_. Learning the kernel bandwidth is a key enabler for scalability and, on its own, allows the method to surpass standard SVGD on challenging benchmarks (e.g., Fig. 6a).
>
> 3. _MH correction (Sec. 3.4)_. The MH step is particularly beneficial in non-smooth settings and in the RL experiments, where it helps preserve early-stage exploration while improving exploitation at later stages.

---

> > ### Author Rebuttal · Reviewer_aqPx · 2026-04-04
> >
> > Thanks a lot for the answers! I still maintain my reservation about approximate accept-reject ratios; controlling errors is might be a step in the right direction, but in the context of Metropolis-Hastings, this is not sufficient to obtain strong convergence guarantees. Overall, I will be maintaining my score.

---

> > > ### Author Response · Authors · 2026-04-04
> > >
> > > Thank you for following up. We are glad we addressed the reviewer’s concerns.
> > >
> > > We would greatly appreciate it if you could elaborate more on what you think is missing for ensuring strong convergence via the MH step augmentation.

---

### Decision · Program_Chairs · 2026-04-30

**Decision:**

Accept (regular)

**Comment:**

This work proposes an MET-SVGD, an extension of the particle-based P-SVGD method of Stein Variational Gradient Descent for the entropy estimation of unnormalized densities, a notoriously challenging problem. By providing better parameter selection and an efficient Hessian estimator, the authors show that MET-SVGD is able to scale to higher-dimensional settings in which P-SVGD struggles.

Reviewers were convinced by the technical contributions of the work, particularly its diagnosis of failure modes in P-SVGD. They also agreed that the problem is highly important. Reviewers disagreed about the experimental evidence provided by the authors, however, with some arguing the results presented were already a strong contribution (by improving over P-SVGD) and others contending that claims of good performance on sampling/estimating entropy from unnormalized densities should feature a much broader set of comparisons with other approaches.

Overall, this is solid work that has positioned itself in a small niche as an improvement over a particular algorithm (P-SVGD). Comparison to a broader range of entropy estimation approaches in high dimensions would broaden its appeal.